# Protocol for the INFORMED (Individualised Patient Care and Treatment for Maternal Diabetes) Study: a randomised controlled trial embedded within routine care

Cassy F Dingena,[1] Anvesha Mahendra,[1] Melvin J Holmes,[1] Naomi S Clement,[2] Eleanor M Scott,[3] Michael A Zulyniak  [1]

[1]Food Science and Nutrition, University of Leeds, Leeds, UK
[2]Medicine, University of Leeds, Leeds, UK
[3]Division of Clinical and Population Sciences, Leeds Institute of Cardiovascular and Metabolic Medicine, University of Leeds, Leeds, UK

**Correspondence to**
Dr Michael A Zulyniak;
m.a.zulyniak@leeds.ac.uk

## ABSTRACT

**Introduction** Diabetes in pregnancy presents a unique physiological challenge to manage glycaemia while maintaining adequate nourishment for the growing fetus. Women with diabetes who become pregnant are at greater risk of adverse maternal and newborn outcomes, compared with women without diabetes. Evidence suggests that control of (postprandial) glycaemia is key to manage maternal and offspring health but it is not yet clear (1) how diet and lifestyle moderate these shifts across the full duration of pregnancy or (2) what aspects of maternal and offspring health are associated with dysglycaemia.

**Methods and analysis** To investigate these gaps, a cross-over randomised clinical trial has been embedded within routine clinical care. Seventy-six pregnant women in their first trimester with type 1 or type 2 diabetes (with or without medication) attending their routine antenatal appointments at National Health Service (NHS) Leeds Teaching Hospitals will be recruited. Following informed consent, data on women's health, glycaemia, pregnancy and delivery will be shared by the NHS with researchers. At each visit in the first (10–12 weeks), second (18–20 weeks) and third (28–34 weeks) trimester, participants will be asked for consent to: (1) lifestyle and diet questionnaires, (2) blood for research purposes and (3) analysis of urine collected at clinical visits. Additionally, participants will be asked to consume two blinded meals in duplicate in second and third trimester. Glycaemia will be assessed by continuous glucose monitoring as part of routine care. The primary outcome is the effect of experimental meals (high vs low protein) on postprandial glycaemia. Secondary outcomes include (1) the association between dysglycaemia and maternal and newborn health, and (2) the association between maternal metabolic profiles in early pregnancy with dysglycaemia in later pregnancy.

**Ethics and dissemination** The Leeds East Research Ethics Committee and NHS (REC: 21/NE/0196) approved the study. Results will be published in peer-reviewed journals and disseminated to participants and the wider public.

**Trial registration number** ISRCTN57579163.

## STRENGTHS AND LIMITATIONS OF THIS STUDY

⇒ The analyses of response to meals alongside repeated measures of blood and urine metabolite profiles will offer insight into distinct shifts in metabolism during pregnancy in women with type 1 and type 2 diabetes, and their association with maternal and newborn health in pregnancy and at delivery.

⇒ The trial is embedded within standard clinical care and uses routine data and biological samples collected by health services to minimise (1) participant burden and (2) non-essential participant contact, which reduces risk of bias and will permit the study to continue even with the re-introduction of public restrictions.

⇒ As with all observational data, participant recall data (sleep, physical activity and diet) are subject to social desirability bias but the inclusion of repeated and complementary measures (ie, metabolite) will allow this to be evaluated in future analysis.

⇒ INFORMED is being conducted with the National Health Service of the UK; therefore, its results may not be directly generalisable to other nations or government health services.

## INTRODUCTION

### Background and scope

Pregnancy naturally induces a state of mild insulin resistance (IR) to shuttle more nutrients to the growing baby; however, in women with diabetes in pregnancy (DIP), excessive IR and persistent hyperglycaemia increases the risk of adverse pregnancy outcomes.[1–4] Globally, the prevalence of DIP is on the rise, affecting ~17% of all pregnancies.[4 5] Compared with women without diabetes, women with DIP are at elevated risk of pre-eclampsia, preterm delivery and mortality, while their offspring are at increased risk of unhealthy weight (<2.5 kg or >4.5 kg), dysglycaemia, injuries at birth,

and higher risk of type 2 diabetes mellitus (T2DM) and cardiovascular disease in later life.[1 5 6]

Postprandial glycaemic control is important for healthy pregnancy outcomes.[1 6] Evidence supports a healthy diet and lifestyle—that includes whole grains, fruits and vegetables, and regular physical activity—as the cornerstone for managing DIP, which is effective in 70%–85% of women with DIP.[7–9] National Institute for Health Care Excellence (NICE) UK guidelines primarily focus on improving carbohydrate quality by including more low glycaemic index (GI) foods as part of a balanced diet including whole grains, fruits and vegetables to manage glycaemia during pregnancy.[7] Although low GI diets do support the management of mean glucose levels, their effect on reducing episodes of hypoglycaemia and hyperglycaemia and ability to reduce maternal and offspring risk of complications is not clearly established.[10] Alternatively, emerging evidence in preclinical and human studies suggests that the amount of maternal protein intake can improve management of dysglycaemia in DIP,[11] but its effect on metabolism and 24-hour dysglycaemia in pregnancy is unknown. Finally, some women find it challenging to consistently follow a balanced diet, due to barriers such as availability, accessibility and affordability of healthy foods, lack of time and cooking skills[7 9] therefore, a cost-effective nutritious meal replacement may be useful for supporting healthy eating habits. Most recently, we noted that 'morning' is when pregnant women with diabetes struggle most to manage glucose levels within a healthy range,[10] suggesting that breakfast may be a particularly important point in the day to offer support for managing dysglycaemia.

Continuous glucose monitors (CGMs) are becoming routinely used in the UK National Health Service (NHS) in perinatal clinical settings for women with DIP.[1] The unobtrusive patches record an individual's glucose every 5 min for up to 14 days and offer quantitative information to identify interstitial glucose deviations over a 24-hour period. By measuring glucose continuously over hours and days, a more complete representation of dysglycaemia can be modelled and offer novel insight regarding the parameters that drive and associate with dysglycemia, and their impact on maternal and offspring health.[11]

Previous studies[12 13] have uncovered new associations and identified novel points of interest for managing dysglycaemia during pregnancy and mediating health risks. However, our ability to inform new strategies to manage these new areas of concern are limited by our understanding of the contribution of biological, lifestyle, and environment exposures on dysglycaemia in early, mid, and late pregnancy and their moderating effect on maternal and offspring health. To address this gap in current understanding, this study aims to investigate the effect of breakfast meal replacements and dietary protein on glucose variability in pregnancy in women with pre-existing type 1 or type 2 diabetes.

## Aim and objectives

Our overall aim is to investigate postprandial CGM profiles throughout the course of the pregnancy and how they are associated with personal (lifestyle) characteristics and physiological parameters. Our primary research objective is to assess the effect of easy-to-prepare meals and dietary protein on dysglycaemia over the course of pregnancy. Secondary research objectives include (1) the association between dysglycaemia and maternal and newborn health, and (2) the association between maternal metabolic profiles in early pregnancy with dysglycaemia in later pregnancy.

## METHODOLOGY AND ANALYSIS
### Participants

Women with type 1 diabetes mellitus (T1DM) and T2DM during pregnancy in their first trimester will be recruited from the DIP antenatal clinics at Leeds Teaching Hospitals NHS Trust (LTHT). Women will be approached by their direct clinical care team and given a study information flyer and invited to contact the research team (via phone or email) if they are interested to participate or if they wish to discuss the study in more detail. Women expressing interest at the end of the initial meeting will be emailed a participant information sheet and a web link to secure electronic informed consent. Once a secure electronic signature is provided, the participant's eligibility will be assessed according to study inclusion and exclusion criteria.

### Sample size using power calculation

CGM data provide numerous metrics to offer unique insight into variations and deviations of glucose levels over time—area under the curve (AUC), mean glucose, coefficient of variation of glucose, mean amplitude of glucose excursions (MAGE) and time in range (TIR).[14] Additionally, there is no current evidence regarding the effect of diet composition and glycaemic load on metrics of CGM in pregnant women with diabetes throughout the duration of pregnancy. Therefore, we have elected to focus on AUC as the primary metric because it is easily interpretable and commonly used to quantify postprandial glycaemia. Evidence from Fabricatore et al[15] demonstrated a significant association ($p < 0.05$) between self-reported GI and measures of CGM (including AUC, mean glucose and % time hyperglycaemic) in a clinical trial of 21 women and 5 men with type 2 diabetes. Assuming similar effect sizes between GI (per unit) and AUC glucose ($\beta = 0.36$ mg/dL/min; $R^2 = 0.38$), mean glucose ($\beta = 0.02$ mol/L; $R^2 = 0.38$) and time spent >10 mol/L blood glucose ($\beta = 0.41\%$; $R^2 = 0.36$), we will have sufficient power (power=0.90) to detect a significant pairwise effect of GI on these parameters with 63 participants. Another study by Law et al suggested that this will provide sufficient power (power=0.90) to compare subgroups of participants (stratified by body mass index (BMI), age, type of diabetes) and test for significant differences (of a minimum effect size) in AUC glucose

**Table 1** Study participant inclusion and exclusion criteria

| Inclusion criteria | Exclusion criteria |
| --- | --- |
| Women aged 18–45 years | Women under 18 or above 45 years of age |
| Singleton pregnancy | Multiple pregnancy |
| Women in the first trimester of pregnancy | Fetal congenital abnormality |
| Previously diagnosed with type 1 or type 2 diabetes mellitus | No diagnosis of diabetes |
| | Diagnosis of gestational diabetes |
| | Significant coexistent medical condition (e.g., overt diabetes complications, cancer, gut mobility or digestion disorder) |
| | Significant psychological (e.g., anorexia, bulimia) and/or mental disorders which undermine informed consent |
| | Dietary allergies or intolerance for the experimental meals |
| | Lack of internet access on a computer or tablet at home |
| | Unable to understand written English and provide informed consent |

($\pm 61\,mmol/L/min$), mean glucose ($\pm 0.5\,mmol/L$) and % time hyperglycaemic ($\pm 3.7\%$).[16] Finally, given the comparable proportions of women reported to be of white European versus non-white European ancestry (57% vs 43%) or diet versus diet+medication (46% vs 54%), we also anticipate having adequate power to compare these confounders of glycaemic response. To account for attrition, we will increase our recruitment target by 20% above our calculated, suggesting a target sample size of 76 recruited women. We have allocated ~6 months to recruit 76 women (at 10–12 weeks' gestation) and ~15 months for study completion (ie, final delivery). All power analyses were performed using G*Power (V.3.1).[17]

### Study participation inclusion and exclusion criteria

All pregnant women over the age of 18 years, with preexisting T1DM or T2DM, in their first trimester and a singleton pregnancy, will be considered for the study. Women who develop diabetes in pregnancy (i.e., gestational diabetes) will not be eligible for the study because they would not be offered CGM until 26–28 weeks' gestation. Exclusion criteria include: (1) inability to understand English sufficiently to read the participant information sheet and provide consent (online supplemental materials A–C); (2) mental and/or psychological disorders that undermine informed consent; (3) cancer, digestive tract disorders; (4) lack of internet access on a computer or tablet at home. Detailed exclusion criteria are shown in table 1 (study screening questionnaires are provided in online supplemental materials D and E).

### Data collection stages and procedures

The study will proceed in three stages at the St James's Hospital (LTHT) with the dietary intervention and interviews conducted remotely (e.g., participant's home) (figure 1). All pregnant women with pre-existing T1DM or T2DM are scheduled for regular NHS clinical visits every 2 weeks throughout the pregnancy. Each woman has an assigned diabetes midwife who caseloads her

pregnancy and liaises with the rest of the clinical care team. Due to COVID-19 and intermittent lockdowns, pregnant women only attend face-to-face meetings at the clinic when due for a scan (*a dating scan* at 10–12 weeks, an *anomaly scan* at 18–20 weeks and *growth scans* at 26–28, 32–34 and 36 weeks of pregnancy) unless deviated due to complications or early delivery. All women with T1DM and T2DM are currently offered CGM as part of their clinical care. The CGM data are automatically uploaded to a secure, remote clinical database, where it can be securely accessed and downloaded by the clinical team and authorised researchers.

The study will require patient consent (1) permitting secure access to routinely collected clinical details regarding maternal and offspring health at each clinical visit and delivery (i.e., height, weight, blood pressure, HbA1c, lipids), CGM data and delivery outcomes with approved study researchers; (2) permitting researchers to use the residual urine from routine clinically collected samples for metabolite analysis; (3) to conduct online and interview questionnaires to assess diet and lifestyle during each trimester at ~10–12, ~18–20, and ~28–34 weeks; and (4) a 10 mL blood sample to be taken with routine clinical bloods at visits ~10–12, ~18–20, and ~28–34 weeks for subsequent metabolomic and genetic analysis. Each participant will be contacted three times for a phone or video chat (participant preference) for ≤30 min each within 2 weeks of each clinical appointment.

### Patient and public involvement

Patients or the public have not been involved in the design of this pilot and feasibility study. However, upon completion of the study, participants will be invited to provide insight and comments regarding the study itself, the burden enrolment and intervention, and identifying other research priorities relevant to the health condition that the researchers can integrate into future studies. They will also be asked if they consent to follow-up discussions

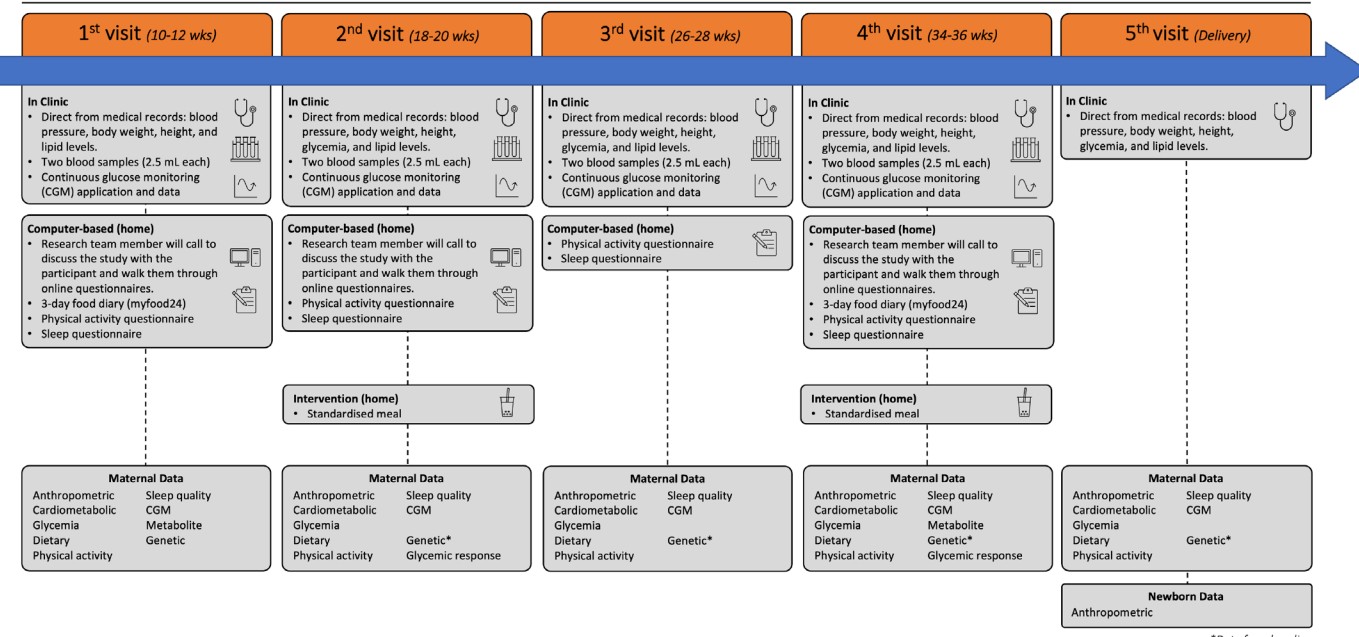

**Figure 1** Study flow chart. Participant will be monitored and in contact with the clinical and research team through the study, starting at 10–12 weeks' gestation. At each clinical visit, routine data will be collected from each participant as standard of care (e.g., anthropometrics, blood samples and CGM). This information will be supplemented with lifestyle information (e.g., diet and sleep) collected directly from the participant via electronic and internet questionnaires. The maternal and offspring data available for analysis at each time point are listed below the timeline. The interventions will be delivered at two time points (18–20 and 34–36 weeks). CGM, continuous glucose monitoring; NHS, National Health Service.

and are keen to be updated on study results and publication material. These points will be invaluable for guiding future work in this area:

*First call between 10 and 12 weeks of pregnancy:* a member of the research team will provide details of the study with the participant and answer any questions. The participants will be instructed to record their dietary intake for 3 days, including 2 weekdays and 1 weekend, using the myFood24 app. The myFood24 is a validated online food diary system created to analyse nutritional intake, which has previously been used in pregnancies complicated by diabetes.[18] Details on the participant's recent physical activity levels (Par-Q for pregnancy)[19] and sleep quality (Leeds Sleep Evaluation)[20] will be recorded by interview (online supplemental materials F and G), while patterns and habitual mealtimes will be collected by myFood24.[21] The dietary meal intervention in the study will also be discussed.

*Second call between 18 and 20 weeks of pregnancy:* participants' compliance in the study and details on physical activity, sleep and habitual mealtimes will be re-recorded. A reminder and summary of when and how to use the myFood24 app will be given. The participant will also be instructed about how and when their study intervention's first set of experimental meals should be consumed.

*Third call between 28 and 34 weeks of pregnancy:* participants' compliance and information transfer as in the second call will be repeated. The participant will also be instructed about how and when their study intervention's second set of experimental meals should be consumed.

Subsequently, the participant will be thanked for their participation in the study and not contacted again after this point.

## Nested cross-over dietary meal intervention within the study

Shortly after their clinical visit at ~18–20 weeks and ~28–34 weeks, participants will be asked to consume standardised breakfast meal replacements in their own home, at breakfast time for 4 days. These will be two different experimental meals (A and B; matched for 400 kcal and 13 g of fat) consumed under free-living conditions. The experimental meals will appear and taste similar but differ in protein quantity which will alter how quickly the glucose in each meal is absorbed into the blood.[22] One experimental meal will have 20 g of vegan and gluten-free protein powder added, which slows gastric emptying and glucose absorption into the blood to a rate that is comparable with commonly consumed whole-grain breakfast cereals (eg, steel-cut or rolled oats; GI ≈40). The other experimental meal will have no added protein. The experimental meals (labelled A or B) with a drink shaker are stored in a box by the research team in a COVID-19-safe laboratory. Each participant will be assigned to one of six random orders to consume the experimental breakfast meals (AABB, ABAB, BBAA, BABA, ABBA, BAAB) over 4 days using an online randomiser (https://www.random.org/integer-sets/). This will be done by block randomisation to assign 12–13 participants to each of the six possible orders of meal consumption, which the participant will follow for both sets of meals (i.e., ~18–20 and

~28–34 weeks). The participant will only need to pour the powder into the shaker, add cold water to the line marked on the cup (500 mL), and consume within 5 min.

The powder is a nutritionally complete meal replacement in drink form; every meal contains a balance of protein, carbohydrates, essential fats, fibre, plus all 26 essential vitamins and minerals, and phytonutrients. Additionally, the product is low in sugar, lactose-free, contains no nuts or palm oil, and has a long shelf life. The powder is commercially available and is produced in facilities that meet highest quality standards. This product was chosen to minimise time burden for participants and it is free from many commonly avoided food items (i.e., lactose, nuts, gluten and meat).

Participants will be asked to avoid consuming other foods and drinks (aside from water) for 2 hours, after which they may consume food as usual. However, they are freely permitted to measure their own blood glucose levels at any point and will be advised to manage any hyperglycaemic or hypoglycaemic events, even if this means eating or drinking within 2 hours of the meal. Participants will be asked to inform the research team of any events by email as soon as possible.

### Data management

The research team will assign unique random screening IDs at the recruitment phase. The study ID will be used to pseudonymise (using personal participant identification numbers) and harmonise the data shared by the NHS clinical database (clinical records) with the data collected by the research team at the University of Leeds (ie, questionnaires, metabolite and genetic data). Only the clinical team and authorised members of the research team will be able to link the study ID to the participant. All personally identifiable information will be stored in a password-protected and encrypted database in a secure area.

*Clinical data:* standard care to measure and collect maternal anthropometric, glycaemic, medication, lipid levels, and blood pressure information during routine hospital visits and during and after labour. Offspring anthropometry measures are taken by the direct healthcare team, which is part of routine care. The women will be asked to give consent for the research team to access their clinical records to obtain these data.

*CGM:* standard clinical care in T1DM and T2DM pregnancies; women will be asked to give consent for the research team to access their CGM data. Of the numerous metrics provided by CGM—that is, AUC, mean glucose, coefficient of variation of glucose, MAGE and TIR—AUC will be the primary metric for the study.

*Urine samples:* standard clinical care; we will ask for up to 2.5 mL of any urine not required for clinical analysis to be saved for research use. These samples will be stored for subsequent metabolic analysis. All samples will be stored at the University of Leeds in designated Human Tissue Act-approved and compliant facilities.

*Blood samples:* standard clinical care requires blood samples for analysis. At the time of routine collection

at each clinical visit, an additional 10 mL blood will be collected for this study. These blood samples will be stored for subsequent metabolic and genomic analysis (relevant to nutrition/diabetes/pregnancy and fetal growth). All samples will be stored at the University of Leeds in designated and secure facilities.

*Lifestyle questionnaires:* at three time points across pregnancy (~10–12, ~18–20 and ~28–34 weeks' gestation), a designated member of the research team will call (phone or video) the participant to complete the questionnaires on physical activity, sleep quality/patterns and habitual mealtime habits.

*Dietary records:* at two time points across pregnancy (~18–20 and ~28–32 weeks of gestation), each participant will be asked to record their diet for 3 days (2 week days, 1 weekend day) using a validated online semiquantitative food frequency questionnaire (myFood24), which estimates dietary intake data (i.e., macronutrients, micronutrients and vitamins for up to 220 nutrients) according to McCance and Widdowson (seventh edition) and branded items that offer nutritional data.[23] Briefly, the data are provided to researchers as a spreadsheet with anonymised identifiers for each participant that can be directly imported into R for analysis. The performance of myFood24 and telephone-based 24-hour dietary recall is in agreement (interclass correlation 0.4–0.5).[23]

### Outcomes of interest

The primary outcome of interest is CGM glucose data, with AUC glucose as the primary CGM metric of interest. Secondary outcomes of interest are associations between metrics of dysglycaemia during pregnancy with maternal outcomes (e.g., gestational weight gain, pre-eclampsia, hypertension, mode of birth, birth trauma, preterm delivery and metabolism) and infant outcomes (e.g., birth weight, height, preterm delivery, mortality, birth trauma, hypoglycaemia, congenital malformation, head and abdominal circumference, perinatal morbidity), and the moderating effects of genetics, metabolism, and diet and lifestyle. All secondary analyses are considered exploratory.

### Statistical analysis considerations

All standard CGM metrics will be calculated, with AUC glucose as the primary CGM metric (mean±SD). The primary analysis will be the effect of dietary protein on AUC glucose for the 24-hour window immediately after each study meal. The analysis will be constructed as pairwise linear model with study meal (0=low protein, 1=high protein) regressed against 24-hour mean AUC glucose and adjusted for study parameters (e.g., randomised meal order) and participant covariates (e.g., maternal age, ethnicity, parity, BMI, gestational age, physical activity and sleep quality). Statistical significance will be set at $p < 0.05$, where a $p < 0.05$ for study meal will suggest a significant effect of dietary protein on 24-hour postprandial AUC glucose. The direction and significance of covariates will be investigated to identify study and participant mediators

of the association. Statistical analysis will be conducted in R studio and SPSS (Ver. 29+).

Secondary research objectives include (1) the association between dysglycaemia and maternal and newborn health, and (2) the association between maternal diet in early pregnancy with dysglycaemia during pregnancy. These analyses will also be performed using regression models adjusted for covariates, with the assessment of early diet on longitudinal changes in dysglycaemia also adjusted for time points of AUC (i.e., mixed-model). The association between early maternal diet and AUC will be performed using three distinct dietary metrics (calculated from myFood24):

1. Overall diet quality

The association between diet quality and dysglycaemia will be assessed using an overall diet quality score.[24] This scoring method has been modified and used previously to assess maternal diet quality in a multiethnic prospective birth cohort.[25] The modified Alternative Healthy Eating Index (mAHEI) score is calculated using the following method; an individual will receive 10 points for each of the following food categories when they consume above or below a threshold of: ≥5 servings of vegetables, ≥4 servings of fruits, ≥1 serving of nuts or soy proteins, ≥3 servings of whole grains, a ratio of ≥4 servings of fish to 1 serving of meat and eggs, and ≤0.5 servings of less-healthy foods (i.e., fried foods and processed meats). Intermediate intake is scored proportionally between 0 and 10. The maximum mAHEI score is 60; the higher the score, the more healthful the participant's diet.

2. Macronutrient

Daily macronutrient consumption (total carbohydrates, proteins and fats) and markers of GI quality (fibre, sugars) will be adjusted for energy and regressed against CGM measures of dysglycaemia. Doing so will allow for the contribution of individual macronutrients on glycaemic measures to be evaluated.

3. Cardinal foods

Partial least squares will be used to identify foods that are more commonly observed in participants with favourable or unfavourable glycaemic control (identified above/below median for glycaemic measures). These foods will then be investigated for their association with measures of dysglycaemia using a regression model.

### Quality control

All participants will receive standard clinical care as per NICE guidance, which will minimise researcher bias. The primary outcome measures are based on laboratory measurements and predetermined cut-off values, which the researchers will not be able to influence. We do not foresee significant researcher bias in collecting antenatal and perinatal outcome data because these will be obtained by clinical staff who are independent of the study outcome and from the participants' medical records. We do not foresee any significant researcher bias in collecting lifestyle records, as standardised and validated questionnaires will be used to obtain this

information. Furthermore, all participants will be asked to consume the two standard meals but the order of their consumption will be randomised. We do not foresee any conflict of interests. The data collected will not be used for informing clinical care decisions of specific cases; all the women will continue with their usual clinical care pathway for the duration of their pregnancy, and all women will be free to terminate their participation in the study at any time with no effect on their quality of care.

### Ethics approval

This study has been reviewed and approved by the Leeds East Research Ethics Committee at the University of Leeds (21/NE/0196).

## DISSEMINATION

There is no formal interim analysis planned except the ongoing evaluation of the recruitment numbers. Results will be disseminated in peer-reviewed scientific journals, conference presentations, and publication on (social) media and in newsletters, to inform the participants and wider public.

## DISCUSSION

The INFORMED clinical trial is double-blinded cross-over randomised clinical trial to evaluate the effect of dietary protein within experimental meals on dysglycaemia in women with pre-existing type 1 or type 2 diabetes. Additionally, we will explore (1) the association between dysglycaemia and maternal and newborn health, and (2) the association between maternal metabolic profiles in early pregnancy with dysglycaemia in later pregnancy, which may provide insights into novel precision therapies for women with DIP.[26] The identification of dietary mediators of glucose variability will aid in the development of more efficacious and appropriate strategies to control glucose levels and minimise maternal and offspring risks in women with DIP.[12 13 26]

**Acknowledgements** The myFood24 app was developed through Medical Research Council funding (grant G110235). myFood24 is now being supported by spinout company Dietary Assessment. Requests to use myFood24 should be made to enquiries@myfood24.org.

**Contributors** CFD—conception, original draft preparation, writing (review and editing), approval of the final manuscript. AM—original draft preparation, writing (review and editing), approval of the final manuscript. MJH—supervision, writing (review and editing), approval of the final manuscript. NSC—writing (review and editing), approval of the final manuscript. EMS—conception, supervision, writing (review and editing), approval of the final manuscript. MAZ—conception, supervision, original draft preparation, writing (review and editing), approval of the final manuscript.

**Funding** This work was supported by Wellcome Trust UK (MAZ; 217446/Z/19/Z) and the University of Leeds (CFD; N/A).

**Competing interests** None declared.

**Patient and public involvement** Patients and/or the public were not involved in the design, or conduct, or reporting, or dissemination plans of this research.

**Patient consent for publication**  Not required.

**Provenance and peer review**  Not commissioned; externally peer reviewed.

**ORCID iD**
Michael A Zulyniak http://orcid.org/0000-0003-4944-5521

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
