## [Reviewer comments · BMJ Open]

ARTICLE DETAILS

TITLE (PROVISIONAL)	Protocol for the INFORMED (Individualised Patient Care and Treatment for Maternal Diabetes): A randomised controlled trial embedded within routine care
AUTHORS	Dingena, Cassy; Mahendra, Anvesha; Holmes, Melvin; Clement, Naomi S; Scott, Eleanor; Zulyniak, Michael

VERSION 1 – REVIEW

REVIEWER	Liu, Yan Ping Peking Union Medical College Hospital, Department of Nutrition
REVIEW RETURNED	25-Jul-2022

GENERAL COMMENTS	This is a promising study, hoping to observe the metabolic characteristics of pre-pregnancy diabetes patients and thus seek for appropriate intervention targets.
---

REVIEWER	Marschner, Simone The University of Sydney, Westmead Applied Research Centre
REVIEW RETURNED	26-Sep-2022

GENERAL COMMENTS	This is an interesting study but I feel that it is quite complex and so specific aims need to be explained and the exact outcomes and how they will be analysed. Perhaps a diagram of the meals being provided and when would be useful to see the design of the study. The primary outcome is : A panel of standard CGM metrics (such as mean glucose, AUC, coefficient of variation and time in/above/below target range) will be calculated as measurements of glycaemic control and presented as mean\pmSD. I would suggest you focus on one as the primary one to assess the primary objective or adjust for multiple comparisons. It needs to clearly stated what the analysis methodology is to answer the primary objective. It mentions many methods but not exactly what will be done to answer the primary objective. There is mention of t-tests and regression tests but more details of the model should be provided to answer the research questions. It is not clear what the primary objective is. What is the primary variable of interest? The meal? Are we testing which meal is better? In the introduction it states: the effect of dietary mediators on (i) 7-day glucose variability in early, mid, and late pregnancy; (ii) and their association with maternal and offspring health during pregnancy and at delivery; and (iii) the moderating effect of
---

	biology, ethnicity, metabolism, and lifestyle factors on glycemia throughout pregnancy affected by Type 1 or Type 2 Diabetes Then we have: Our primary research objective is to assess the role of diet as a mediator of dysglycemia over the course of pregnancy. Then in the discussion it says: I to evaluate the safety and efficacy of experimental meals that differ in rate of glucose absorption on glucose variability during pregnancy (using CGM) I am confused what the primary focus is. The last sentence of the abstract should state the aim not the design.
--	--

REVIEWER	Zoubeidi , taoufik United Arab Emirates University, Statistics
REVIEW RETURNED	15-Oct-2022

GENERAL COMMENTS	Remarks about the article Protocol for the INFORMED (Individualised Patient Care and Treatment for Maternal Diabetes): A randomised controlled trial embedded within routine care The proposed clinical trial aims to investigate the effects of diet on dysglycemia in T1DM and T2DM pregnant women and the mediating effects of the subject's personal characteristics on dysglycemia. The study protocol is a double-blind, cross-over randomised clinical trial which is embedded within routine clinical care of pregnant women. The study is well designed. The validity of the intended measurements is adequate. Moreover, most of the measurements to be collected are part of the routine clinical care of pregnant women which reduces the burden on participating subjects and prevents potential data collection biases. I have the following remarks/comments on the protocol. 1- Page 5, first paragraph. I could not find the bibliographic reference of Law et al. (2019) in the References section of the document. Moreover, the document does not provide sufficient information on how the power of 0.9 was obtained. If R was used to compute the powers listed in this paragraph, then it would be informative to specify which R package was used or else provide a reference for the method used to compute the powers. The symbol β is used in this paragraph to represent both the regression coefficients and the type II error, i.e., power = 1 - β. Use different symbols to represents these two parameters. 2- The 76 participants will be randomly assigned to one of six sequences of breakfast meals (AABB, ABAB, BBAA, BABA, ABBA, BAAB) over 4 days using an online randomiser. It is not clear whether the assignments will be completely random or whether the assignment will be done randomly while ensuring that similar numbers of subjects are assigned to each sequence, that is, the assignment will ensure that about $76/6 \approx 12$ patients are assigned to each sequence. If the assignment is done completely at random, there is a possibility that one or more sequences will be little or not represented in the study. 3- Page 9, line 34. In this paragraph and other sections in the document state that three interviews (online survey) will be carried
---

	out at ~10-12 weeks, ~18-20 weeks, and ~28-34 weeks. However, in the chart on page 15, the online survey is done 4 times. 4- Page 9, line 45. At two timepoints each participant will be asked to record her diet for 3 days (2 weekdays, 1 weekend day). The possibility of a Social Desirability Bias occurring under this approach needs to be mentioned in the limitations. Such bias arises when the participants may be more careful in choosing what they consume when they are observed by an interviewer. Another interviewing approach is a dietary recall where patients are asked to recall what they consumed in previous days. While not necessarily better than the proposed approach, it is worth mentioning why the proposed approach is preferred. 5- The study data is longitudinal with at least 5 timepoints (visits). Will the analysis use a longitudinal statistical model (mixed models) to account for the change over time per subject or will separate analyses be done at each timepoint (visit)? 6- Page 5, line 28. Replace "or" by "our" in the phrase "we will increase or recruitment target..." 7- Page 8, line 3. Add "a" before "rate" in the phrase "...which slow gastric emptying and glucose absorption into the blood to rate that is comparable to commonly consumed whole grain breakfast cereals..." 8- Page 8, line 41. Correct the sentence "We will ask that they inform the research team of any events by email as soon as possible and these will be discussed any follow up."
--	---

REVIEWER	Greiner, Gregory Deutsches Diabetes-Zentrum Leibniz-Zentrum für Diabetes-Forschung, Institute for Health Services Research and Health Economics, German Diabetes Center
REVIEW RETURNED	08-Dec-2022

GENERAL COMMENTS	the study protocol is well written. The attached materials (study information, questionnaires, etc.) complement the content of the protocol and thus provide a comprehensive insight into the study. The aim of the study is clearly described, as are the intended methods. I think that some minor additions and further information could be helpful for the protocol, here are my suggestions::  1. A brief description of the rationale for excluding (non-including) women with gestational diabetes. 2. It is not entirely clear how the data will be processed by the MyFood app or made available to the research team. 3. In the limitations you can describe how the research team assesses the reporting bias using the MyFood app, especially against the background of social desirability (esp. in pregnancy) and how it compares to a questionnaire collected by telephone. 4. For comparability of the studies, it would be helpful to cite the sources of the questionnaires and materials used. 5. The research team could describe why they chose only experimental breakfasts and not also (or only) other mealtimes. As a reader, one wonders whether this would not provide deeper insights into mealtime glucose responses.
---

VERSION 1 – AUTHOR RESPONSE

Reviewer: 1

Dr. Yan Ping Liu, Peking Union Medical College Hospital

Comments to the Author:

This is a promising study, hoping to observe the metabolic characteristics of pre-pregnancy diabetes patients and thus seek for appropriate intervention targets.

We thank Reviewer 1 for their support of the INFORMED study.

Reviewer: 2

Dr. Simone Marschner, The University of Sydney

Comments to the Author:

This is an interesting study but I feel that it is quite complex and so specific aims need to be explained and the exact outcomes and how they will be analysed. Perhaps a diagram of the meals being provided and when would be useful to see the design of the study.

We thank the reviewer for their comment and have revised our study timeline diagram (Fig 1) to more clearly show when the meals will be delivered.

The primary outcome is : A panel of standard CGM metrics (such as mean glucose, AUC, coefficient of variation and time in/above/below target range) will be calculated as measurements of glycaemic control and presented as mean \pm SD. I would suggest you focus on one as the primary one to assess the primary objective or adjust for multiple comparisons.

We thank the reviewer for noting this and agree that electing a primary metric is best. We have listed Area Under the Curve as our primary metric for glycemia because it is routinely used in both CGM and non-CGM studies and is an effective measure for quantifying the magnitude of deviation of blood glucose from baseline. The other metrics will be recorded and assessed as secondary measures. We have modified the text in the manuscript to reflect this: "Of the numerous metrics provided by CGM – i.e., area under the curve (AUC), mean glucose, coefficient of variation (CV) of glucose, mean amplitude of glucose excursions (MAGE), and time in range (TIR) – AUC will be the primary metric for the study."

It needs to clearly stated what the analysis methodology is to answer the primary objective. It mentions many methods but not exactly what will be done to answer the primary objective. There is

mention of t-tests and regression tests but more details of the model should be provided to answer the research questions.

With your recommendation to elect a primary CGM metric (i.e., AUC), we have revised the statistical methods section to more clearly explain the assessment of this outcome in detail, with additional information for secondary outcomes. The section now reads: "All standard CGM metrics will be calculated, with AUC glucose as the primary CGM metric (mean±SD). The primary analysis will be the effect of dietary protein on AUC glucose for the 24-hr window between (or following) each study meal. The analysis will be constructed as pairwise linear model with study meal (0 = low protein, 1 = high protein) regressed against 24-hr mean AUC glucose and adjusted for study parameters (e.g., randomised meal order) and participant covariates (e.g., maternal age, ethnicity, parity, BMI, gestational age, physical activity, and sleep quality). Statistical significance will be set at $p < 0.05$, where a $p < 0.05$ for study meal will suggest a significant effect of dietary protein on 24-hr postprandial AUC glucose. The direction and significance of covariates will be investigated to identify study and participant mediators of the association. Statistical analysis will be conducted in R studio and SPSS."

It is not clear what the primary objective is. What is the primary variable of interest? The meal? Are we testing which meal is better? In the introduction it states: the effect of dietary mediators on (i) 7-day glucose variability in early, mid, and late pregnancy; (ii) and their association with maternal and offspring health during pregnancy and at delivery; and (iii) the moderating effect of biology, ethnicity, metabolism, and lifestyle factors on glycemia throughout pregnancy affected by Type 1 or Type 2 Diabetes. Then we have: Our primary research objective is to assess the role of diet as a mediator of dysglycemia over the course of pregnancy. Then in the discussion it says: I to evaluate the safety and efficacy of experimental meals that differ in rate of glucose absorption on glucose variability during pregnancy (using CGM). I am confused what the primary focus is.

Our apologies for the confusion and agree that this was not adequately clear. With the comments addressed above and the accompanying revisions in introduction, methods, and discussion we feel that this is now clear. For example, our final sentence of the introduction now reads: "this study aims to investigate the effect of breakfast meal replacements and dietary protein on glucose variability in pregnancy in women with pre-existing Type 1 or Type 2 Diabetes". Additionally, the 'Aim and Objective' section has been revised to read: "Our primary research objective is to assess the effect of easy to prepare meals and dietary protein on dysglycemia over the course of pregnancy. Secondary research objectives include (i) the association between dysglycemia and maternal and newborn health, and (ii) the association between maternal metabolic profiles in early pregnancy with dysglycemia in later pregnancy."

The last sentence of the abstract should state the aim not the design.

We revised the abstract and feel that it now reads more clearly.

Reviewer: 3

Prof. taoufik Zoubeidi , United Arab Emirates University

Comments to the Author:

Remarks about the article Protocol for the INFORMED (Individualised Patient Care and Treatment for Maternal Diabetes): A randomised controlled trial embedded within routine care

The proposed clinical trial aims to investigate the effects of diet on dysglycemia in T1DM and T2DM pregnant women and the mediating effects of the subject's personal characteristics on dysglycemia. The study protocol is a double-blind, cross-over randomised clinical trial which is embedded within routine clinical care of pregnant women.

The study is well designed. The validity of the intended measurements is adequate. Moreover, most of the measurements to be collected are part of the routine clinical care of pregnant women which reduces the burden on participating subjects and prevents potential data collection biases.

I have the following remarks/comments on the protocol.

1- Page 5, first paragraph. I could not find the bibliographic reference of Law et al. (2019) in the References section of the document. Moreover, the document does not provide sufficient information on how the power of 0.9 was obtained. If R was used to compute the powers listed in this paragraph, then it would be informative to specify which R package was used or else provide a reference for the method used to compute the powers. The symbol β is used in this paragraph to represent both the regression coefficients and the type II error, i.e., power = 1 - β . Use different symbols to represent these two parameters.

*We thank the reviewer for noting these and have added in the missing reference for Law 2019 (now #15), replaced the equation '1 - β ' with the word 'power', and have added details regarding the software used for the power analysis, specifically: "All power analysis were performed using G*Power (v3.1)¹⁶."*

2- The 76 participants will be randomly assigned to one of six sequences of breakfast meals (AABB, ABAB, BBAA, BABA, ABBA, BAAB) over 4 days using an online randomiser. It is not clear whether the assignments will be completely random or whether the assignment will be done randomly while ensuring that similar numbers of subjects are assigned to each sequence, that is, the assignment will ensure that about 76/6 \approx 12 patients are assigned to each sequence. If the assignment is done completely at random, there is a possibility that one or more sequences will be little or not represented in the study.

We thank the reviewer for the opportunity to expand on the randomisation process and have added the following text to the manuscript to confirm that we will use a block randomization approach: "This will be done by block randomization to assign 12-13 participants to each of the six possible orders of meal consumption, which the participant will follow for both sets of meals (i.e., ~18-20 week and ~28-34 wks)."

3- Page 9, line 34. In this paragraph and other sections in the document state that three interviews (online survey) will be carried out at ~10-12 weeks, ~18-20 weeks, and ~28-34 weeks. However, in the chart on page 15, the online survey is done 4 times.

Our apologies for this error and we thank the reviewer for catching this. The text was correct, the online survey will be done 3 times. The figure has been corrected.

4- Page 9, line 45. At two timepoints each participant will be asked to record her diet for 3 days (2 weekdays, 1 weekend day). The possibility of a Social Desirability Bias occurring under this approach needs to be mentioned in the limitations. Such bias arises when the participants may be more careful in choosing what they consume when they are observed by an interviewer. Another interviewing approach is a dietary recall where patients are asked to recall what they consumed in previous days. While not necessarily better than the proposed approach, it is worth mentioning why the proposed approach is preferred.

This is a valid point and we have added the following text. In the 'Data Management' section, we now cite and present the validation study that reported interclass correlation between myFood24 and 24-hour telephone recall (ICC~0.4-0.5), while in the 'Strengths and Limitations' section: "As with all observational data, participant recall data (sleep, physical activity, and diet) are subject to social desirability bias but the inclusion of repeated and complementary measures (i.e., metabolite) may allow this to be evaluated in future analysis."

5- The study data is longitudinal with at least 5 timepoints (visits). Will the analysis use a longitudinal statistical model (mixed models) to account for the change over time per subject or will separate analyses be done at each timepoint (visit)?

We thank the reviewer for noting this limited explanation. In brief, our primary focus will be the 24-hr period immediately after consuming the study meal but we will also investigate longitudinal changes. To explain this clearly, the statistical section now reads: "All standard CGM metrics will be calculated, with AUC glucose as the primary CGM metric (mean±SD). The primary analysis will be the effect of dietary protein on AUC glucose for the 24-hr window immediately after each study meal. The analysis will be constructed as pairwise linear model with study meal (0 = low protein, 1 = high protein) regressed against 24-hr mean AUC glucose and adjusted for study parameters (e.g., randomised meal order) and participant covariates (e.g., maternal age, ethnicity, parity, BMI, gestational age, physical activity, and sleep quality). Statistical significance will be set at $p < 0.05$, where a $p < 0.05$ for study meal will suggest a significant effect of dietary protein on 24-hr postprandial AUC glucose. The direction and significance of covariates will be investigated to identify study and participant mediators of the association. Statistical analysis will be conducted in R studio and SPSS.

Secondary research objectives include (i) the association between dysglycemia and maternal and newborn health, and (ii) the association between maternal diet in early pregnancy with dysglycemia during pregnancy. These analyses will also be performed using regression models adjusted for covariates, with the assessment of early diet on longitudinal changes in dysglycemia also adjusted for timepoints of AUC (i.e., mixed-model)."

6- Page 5, line 28. Replace "or" by "our" in the phrase "we will increase or recruitment target..."

Thank you, this has been corrected.

7- Page 8, line 3. Add “a” before “rate” in the phrase “...which slow gastric emptying and glucose absorption into the blood to rate that is comparable to commonly consumed whole grain breakfast cereals...”

Thank you, this has been corrected.

8- Page 8, line 41. Correct the sentence “We will ask that they inform the research team of any events by email as soon as possible and these will be discussed any follow up.”

Thank you, the sentence has been revised to reads more clearly, written: “Participants will be asked to inform the research team of any events by email as soon as possible.”

Reviewer: 4

Mr. Gregory Greiner, Deutsches Diabetes-Zentrum Leibniz-Zentrum für Diabetes-Forschung,
Universitätsklinikum Düsseldorf Centre for Health and Society

Comments to the Author:

Dear Authors,

Dear Editors,

the study protocol is well written. The attached materials (study information, questionnaires, etc.) complement the content of the protocol and thus provide a comprehensive insight into the study. The aim of the study is clearly described, as are the intended methods. I think that some minor additions and further information could be helpful for the protocol, here are my suggestions:

1. A brief description of the rationale for excluding (non-including) women with gestational diabetes.

Thank you for this recommendation, we’ve added the following text to the inclusion and exclusion description: “Women without T1DM or T2DM that develop gestational diabetes will not be eligible for the study because they would not be diagnosed until 26-28 weeks gestation are not currently offered CGM”.

2. It is not entirely clear how the data will be processed by the MyFood app or made available to the research team.

We thank the reviewer for offering the opportunity to offer more detail. The ‘Dietary Records’ section now reads: “At two timepoints across pregnancy (~18-20 and ~28-32 weeks of gestation), each participant will be asked to record their diet for 3 days (2 week days, 1 weekend day) using a validated online semi-quantitative food frequency questionnaire (myFood24), which estimates dietary

intake data (i.e., macronutrients, micronutrients and vitamins for up to 220 nutrients) according to McCance and Widdowson (7th edition) and branded items that offer nutritional data¹⁹. Briefly, the data is provided to researchers as a spreadsheet with anonymised identifiers for each participant that can be directly imported into R for analysis. The performance of myFood24 and 24 hour dietary recall are in agreement (interclass correlation 0.4-0.5).¹⁹

3. In the limitations you can describe how the research team assesses the reporting bias using the MyFood app, especially against the background of social desirability (esp. in pregnancy) and how it compares to a questionnaire collected by telephone.

Thank you, we now have commented on the issue with bias in the limitation section – “As with all observational data, participant provided data (sleep, physical activity, and diet) are subject to social desirability bias but we the inclusion of repeated and complementary measures (i.e., metabolite) may allow this to be evaluated and minimised in the analysis.”. Additionally, we now report the interclass correlation (ICC) between myFood24 and telephone based 24-hr recall in the main text: “The performance of myFood24 and telephone-based 24 hour dietary recall are in agreement (interclass correlation 0.4-0.5).¹⁹”

4. For comparability of the studies, it would be helpful to cite the sources of the questionnaires and materials used.

Thank you, we have added the questionnaire names and citations to the text: “Details on participant’s recent physical activity levels (Par-Q for pregnancy)¹⁹ and sleep quality (Leeds Sleep Evaluation)²⁰ will be recorded by interview, while patterns and habitual mealtimes will be collected by myFood24²¹.”

5. The research team could describe why they chose only experimental breakfasts and not also (or only) other mealtimes. As a reader, one wonders whether this would not provide deeper insights into mealtime glucose responses.

This is a very important point and one that we appreciate the opportunity to address. Briefly, our recent work demonstrates that while pregnant women with diabetes often struggle to manage glucose levels across the entire day, the morning is when dysglycemia is more severe. This paper was only recently uploaded to MedRxiv and is now cited in the introduction: “Most recently, we noted that ‘morning’ is when pregnant women with diabetes struggle most to manage glucose levels within a healthy range,¹⁰ suggesting that breakfast may be a particularly important point in the day to offer support for managing dysglycemia.”

Reviewer: 1

Competing interests of Reviewer: There's no competing interests

Reviewer: 2

Competing interests of Reviewer: None

Reviewer: 3

Competing interests of Reviewer: Yes

Reviewer: 4

Competing interests of Reviewer: Non